# Restoration of Heel–Toe Gait Patterns for the Prevention of Asymmetrical Hip Internal Rotation in Patients with Unilateral Spastic Cerebral Palsy

**DOI:** 10.3390/children8090773

**Published:** 2021-09-02

**Authors:** Reinald Brunner, William R. Taylor, Rosa M. S. Visscher

**Affiliations:** 1Orthopaedic Department, Children’s University Hospital, Spitalstrasse 33, 4053 Basel, Switzerland; 2Department of Biomedical Engineering, University of Basel, Gewerbestrasse 14, 4123 Basel, Switzerland; rosa.visscher@hest.ethz.ch; 3Laboratory for Movement Biomechanics, Institute for Biomechanics, ETH Zurich, Leopold-Ruzicka-weg 4, 8093 Zurich, Switzerland; bt@ethz.ch

**Keywords:** hip internal rotation, unilateral spastic cerebral palsy, ankle foot orthosis, toe-walking, heel–toe gait

## Abstract

Forward modelling has indicated hip internal rotation as a secondary physical effect to plantar flexion under load. It could therefore be of interest to focus the treatment for patients with unilateral spastic cerebral palsy on achieving a heel–toe gait pattern, to prevent development of asymmetrical hip internal rotation. The aim of this preliminary retrospective cohort investigation was to evaluate the effect of restoring heel–toe gait, through use of functional orthoses, on passive hip internal rotation. In this study, the affected foot was kept in an anatomically correct position, aligned to the leg and the gait direction. In case of gastrosoleus shortness, a heel raise was attached to compensate for the equinus and yet to provide heel–floor contact (mean equinus = −2.6 degrees of dorsiflexion). Differences in passive hip internal rotation between the two sides were clinically assessed while the hip was extended. Two groups were formed according to the achieved correction of their gait patterns through orthotic care: patients with a heel-toe gait (with anterograde rocking) who wore the orthosis typically for at least eight hours per day for at least a year, or patients with toe-walking (with retrograde rocking) in spite of wearing the orthosis who used the orthosis less in most cases. A Student’s *t*-test was used to compare the values of clinically assessed passive hip rotation (*p* < 0.05) between the groups and the effect size (Hedges’ g) was estimated. Of the 70 study participants, 56 (mean age 11.5 y, majority GMFCS 1, similar severity of pathology) achieved a heel-toe gait, while 14 remained as toe-walkers. While patients with heel–toe gait patterns showed an almost symmetrical passive hip internal rotation (difference +1.5 degrees, standard deviation 9.6 degrees), patients who kept toe-walking had an increased asymmetrical passive hip internal rotation (difference +10.4 degrees, standard deviation 7.5 degrees; *p* = 0.001, Hedges’s g = 0.931). Our clinical findings are in line with the indications from forward modelling that treating the biomechanical problem might prevent development of a secondary deformity. Further prospective studies are needed to verify the presented hypothesis.

## 1. Introduction

With an incidence estimated to range between 1.5 and 3.0 per 1000 live births, cerebral palsy (CP) is one of the most frequent causes of motor disability in children [1]. CP refers to a group of permanent, but not unchanging, disorders of motor function affecting movement and posture, which are due to a non-progressive interference, lesion, or abnormality of the developing/immature brain [2]. Patients with CP can be divided into three major groups according to the type of motor problems: spastic, dyskinetic, and ataxic. The spastic type is characterized by enhanced muscle tension, hyperreflexia, and pathological reflexes and can be further categorised into unilateral (hemiplegic) and bilateral (diplegia, tetraplegia) involvement [3]. The degree of motor problems varies from mild (gross motor function classification Scale—GMFCS-I and II, capable of walking independently) to very severe (GMFCS V, fully dependent on caretaker) [4]. In addition to motor problems, children with CP also experience sensory impairments. The dependency of motor function on sensory input may contribute to exacerbating motor deficits on the affected side in unilateral CP [5,6].

Toe-walking has been estimated to occur in over 50% of children with spastic CP [7]. In spastic unilateral CP the natural history of the affected feet shows rather age typical function during gait at an early age, but an involuntary toe-walking pattern tends to develop during later childhood [5]. The common treatment regimen for these children involves orthotics, botulinum toxin injections, and physiotherapy, all aimed to prevent (further) gastrosoleus shortening and maintain or achieve a heel-strike while walking [8,9,10,11]. Later in childhood, patients with unilateral CP often show hip internal rotation, flexion, and adduction problems as well as a knee flexion deformity of the affected leg: the typical hemiplegic posture [12].

The reason for the typical hemiplegic posture is seen in spasticity of the proximal muscles and involving hyperexcitable reflexes in the ankle plantar flexors [13,14]. Spasticity found in clinical exam is expected to occur similarly during function. Soft tissue surgery and bony corrections are generally carried out at all necessary levels. However, other authors did not find evidence that spasticity contributes to toe-walking [15] when sensory feedback to ankle plantar flexors was studied. A possible explanation for this contradiction may be that spasticity during function is overestimated and mechanical factors are involved. Previous work has shown that, in contrast to typical gait, the knee marker trajectory in toe-walkers shows a backward movement during the stance phase of gait [16]. Forward modelling studies revealed that the plantar flexor push under load is a possible cause for this backward trajectory, while at the same time the hip flexes, internally rotates, and adducts [17]. It was hypothesized that the leg is pushed backwards when rocking on the foot is retrograde (toe-heel). In conclusion, plantar flexor overactivity alone could explain the full clinical presentation of the hemiparetic posture. In toe-walkers, initial toe contact leads to an abrupt and large stretch of the Achilles tendon and the attached gastrosoleus. During clinical examination, a spastic reaction in the gastrosoleus is often observed in toe-walking patients with CP. A similar reaction is observed during gait at initial toe contact, as the plantar flexor moment increases steeply after initial toe contact (Figure 1a). Rocking over the heel in more normal gait patterns avoids this stretch at initial contact and the increase of the plantar flexor moment is reduced (Figure 1b).

Gait analysis in spastic unilateral (hemiplegic) CP usually shows the combination of increased plantar flexion with pelvic retraction and hip internal rotation. Often, pelvic retraction is regarded as a compensation for hip internal rotation. A recent forward modelling study, however, indicated that both could be simple physical effects of plantar flexion under load [16]. Based on this effect of plantar flexion under load and the push back of the leg, the treatment concept of patients with unilateral CP could be reconsidered: Instead of focusing on gastrosoleus length to avoid contractures, an initial heel contact could be identified as the primary goal. The present preliminary retrospective cohort study therefore investigated the effect of this new concept on clinically assessed asymmetry of passive hip rotation after patients received orthotic care (to restore heel–toe gait) for at least a year.

## 2. Materials and Methods

### 2.1. Treatment Concept of Local Hospital

As a treatment concept, an ankle foot orthosis (AFO) with free dorsiflexion, but locked plantarflexion worn during function, was fitted for each patient with unilateral CP who started to present signs of toe walking. Leg length was equalized by a heel raise on the contralateral side if required. Toddlers wore the device for only a few hours per day, but as soon as the children started school, the orthosis was worn for at least eight hours per day. In addition, continued physiotherapy was administered in all patients. Treatment continued until the patient either developed a heel–toe gait pattern or a surgical correction of the short Achilles tendon was carried out.

Specific attention was given to the biomechanical construction of the AFO. The treatment aimed at aligning the foot to the thigh and direction of gait: with varus/valgus heel pads, pads under the midfoot, and pads under the forefoot in case of a supination deformity. The foot skeleton was held in the normal anatomical arrangement with the foot in equinus position if required by gastrosoleus contracture. A heel raise compensated for ankle plantar-flexion as much as was required to achieve a full heel contact when standing (Figure 2). A heel rise was added on the contralateral side if required to compensate for any leg length inequality.

Application of a casting to reduce gastrosoleus contracture was avoided, if possible, as it is thought to lead to atrophy and recurrence [18]. In the best case, patients walk with less equinus but with a drop foot due to weakness of the foot dorsiflexors (initial toe contact, but heel contact achieved during mid-stance). Similarly, injections of botulinum toxin were avoided where possible, as it is known to damage the muscle [19,20] and has only a temporary effect on muscle overactivity. The drop foot problem is the same as with casting, and a heel–toe gait pattern would most probably not be achieved.

The study was approved by the local ethical committee (EKNZ 2021-00995).

### 2.2. Patients and Parameters

Patients were included if they visited the local hospital in 2017 or later, were at least four years old at time of assessment, presented with unilateral spastic (hemiplegic) CP without additional neurological or orthopaedic conditions (such as epilepsy, behavioural disorders, hip reconstruction), underwent orthotic treatment for at least a year at the local hospital, and underwent clinical assessment of their gait patterns and passive hip rotation during their visit to the local hospital (lacking in seven patients). Four patients were excluded because general consent was not provided. In addition, participants who received surgery prior to assessment or botulinum toxin injection within six months of assessment were excluded.

All included patients were diagnosed with spastic unilateral (hemiparetic) CP by a paediatric neurologist. The gait pattern of all patients was clinically assessed by a neuro- orthopedic surgeon who was trained in evaluating gait patterns. Patients were grouped into one of two groups based on the gait pattern: Patients who presented a heel–toe gait pattern were grouped in group-H. They wore their orthosis frequently (at least eight hours per day); Patients that presented toe-walking were grouped in group-T. They either did not achieve a heel–toe gait pattern with the orthosis or they wore their orthosis for less time if at all. The groups did not differ considering age, sex, severity of pathology, and type of orthosis (Table 1). The patients were on average 11.5 years old and the majority were classified as GMFCS level 1 (GMFCS I). For 54/56 in Group-H and 11/14 in Group-T, passive internal hip rotation was clinically assessed as well as their level as spasticity (using the modified Ashworth scale [21]), prior to performing clinical gait analysis by a trained physiotherapist. For the other 2/56 in group-H and 3/14 in Group-T, passive internal hip rotation was clinically assessed during their regular clinical check-up with their treating neuro-orthopaedic physician; however, for these patients, spasticity assessment was lacking.

The parameters of interest were: (1) the difference between the affected versus the non-affected side of clinically assessed passive hip internal rotation with the hips in extension; (2) whether a heel-toe gait was achieved during gait while wearing the orthosis, and (3) whether the orthosis was used regularly as indicated by the patient, the parents, and the physiotherapists. These clinical parameters were recorded during the patients’ visits at least once a year.

Most of the patients were fitted with the AFO as described, however, 4/56 patients in Group-H and 2/14 in Group-T were fitted only with a foot orthosis (FO) extending to the malleoli and an adaptive heel raise.

### 2.3. Statistics

Normal distribution of the data was checked with a Shapiro–Wilks test (*p* > 0.25). A two-tailed heteroscedastic Student’s *t*-test (Excel 2010) was applied to check for significance of the data on passive hip rotation between the two groups (*p* < 0.05). Hedges’ g was calculated to estimate the corresponding effect size [22].

## 3. Results

A total of 70 patients were retrospectively included for this cohort investigation. Group-H consisted of 56 patients, Group-T of 14. Age did not significantly differ between the groups. Equinus deformity and spasticity (modified Ashworth scale) were similar across the two groups. A statistically significant difference in passive hip internal rotation difference between the affected and unaffected sides was detected between the groups (Table 1). Hip internal rotation was almost symmetrical in Group-H (mean difference between sides 1.5 degrees), while an asymmetry was found in Group-T (mean difference between sides 10.4 degrees, *p* = 0.001, Figure 3), with Hedges’ g indicating a strong effect size (g = 0.931, Table 1).

As the contracture of the gastrosoleus was not the aim of treatment, patients developed an equinus deformity of 2.6 degrees on average (up to 30 degrees), similar in both groups (Table 1).

## 4. Discussion

Patients with unilateral spastic cerebral palsy (CP) usually present with an equinus, hip internal rotation deformity, and pelvic retraction. Previous work has indicated that proximal deformities could be secondary to foot malfunction [3], plausibly a simple result of form following function. However, such data suggests that treatment of toe-walking patients with unilateral spastic CP could be reconsidered, and better focused on gastrosoleus length treatment aiming to restore heel-toe gait. The present preliminary retrospective cohort investigation indicated that patients who present with a heel–toe gait pattern while walking with an orthosis yield a more symmetrical clinically assessed passive hip internal rotation: bilateral heel–toe gait patterns were accompanied by a more symmetrical passive hip internal rotation in contrast to excessive asymmetrical passive hip internal rotation in unilateral toe-walkers.

As correcting dorsiflexion was not the primary goal, patients generally developed an equinus deformity of some 15 to 25 degrees. As the foot was kept aligned within the orthosis, additional foot deformities requiring correction were rare. In general, orthotic treatment was started when the patients began to walk on their toes. Some patients, however, were seen for the first time only at a more mature age when foot deformities were already present. The effect of late onset of treatment on foot deformities, however, could not be studied due to lack of data in the present cohort. When patients reach puberty, we usually offer a surgical correction to replace the orthosis. While in patients with heel–toe gait patterns, a simple rebalancing of muscle-tendon lengths at the foot is sufficient (we perform an Achilles tendon lengthening in combination with a shortening of the tibialis anterior tendon [23,24]), toe-walkers who developed a clinically assessed asymmetrical passive hip internal rotation often require an additional femoral correction osteotomy, a much more invasive procedure. If the orthotic treatment started late, the foot may have developed an abducted flat-foot deformity which would be reconstructed in the same surgical session. The present study indicates that besides the cosmetic effect of a normal gait pattern, the achievement of a heel–toe gait might also have a preventive effect on the development of secondary deformities at the hip; however, an intervention study needs to be conducted to test any causality suggested by these preliminary indications.

Movement in a joint is defined by bony and soft tissue structures. For hip rotation, we commonly consider femoral torsion as the main parameter for joint movement, but tension within the capsule and ligaments, as well as the passive muscle lengths might also play important roles. The present preliminary retrospective cohort investigation showed that the development of passive hip internal rotation may have a clear interdependency with foot rocking in patients with unilateral CP. Our data suggest that symmetry of passive hip internal rotation (while the hip is in extension) may serve as a clinical biomarker for the ability of correcting toe-walking. The question of the affected structures and the necessary time for the adaptation remains open, as does the question how functional hip internal rotation during gait is influenced by restoring heel–toe gait. Further investigations are required before our understanding of the possible benefit of restoring heel–toe gait can be improved.

This study had a number of limitations: Firstly, we present a retrospective study design, whereby the essential parameters of interest were recorded during regular clinical visits. Unfortunately, the beginning of the orthotic treatment according to the treatment principle was not registered. Secondly, there was no untreated or differently treated control group. However, with the large effect found within this investigation, leaving patients without treatment, and risking major corrective surgery, was considered unethical. Finally, some patients developed a mild equinus deformity during the course of treatment, but foot deformities were not assessed in detail as this was outside of the scope of this preliminary retrospective cohort investigation.

## 5. Conclusions

With this communication, we highlighted that patients with spastic unilateral cerebral palsy who walk with a heel–toe gait pattern with anterograde foot rocking, achieved through use of functional orthoses, yield more symmetrical clinically assessed passive hip rotation compared to those who maintained toe-walking with retrograde rocking. We therefore propose that future prospective studies further investigate the benefits of functional treatment on biomechanical links as a priority over the (secondary) deformity.

## Figures and Tables

**Figure 1 children-08-00773-f001:**
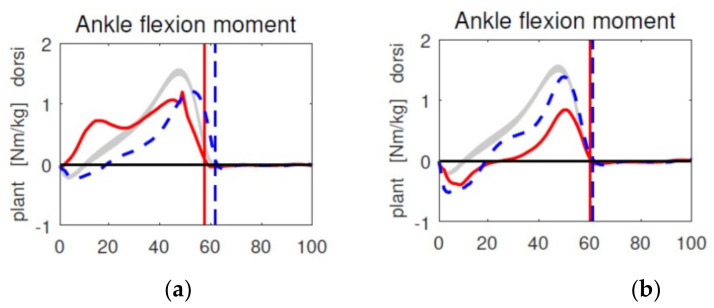
Demonstration of the steep increase of the plantar flexor moment during toe-walking gait (**a**) and the reduced moment when walking with an orthosis (**b**); affected side left (red), unaffected side dotted blue, normal range grey.

**Figure 2 children-08-00773-f002:**
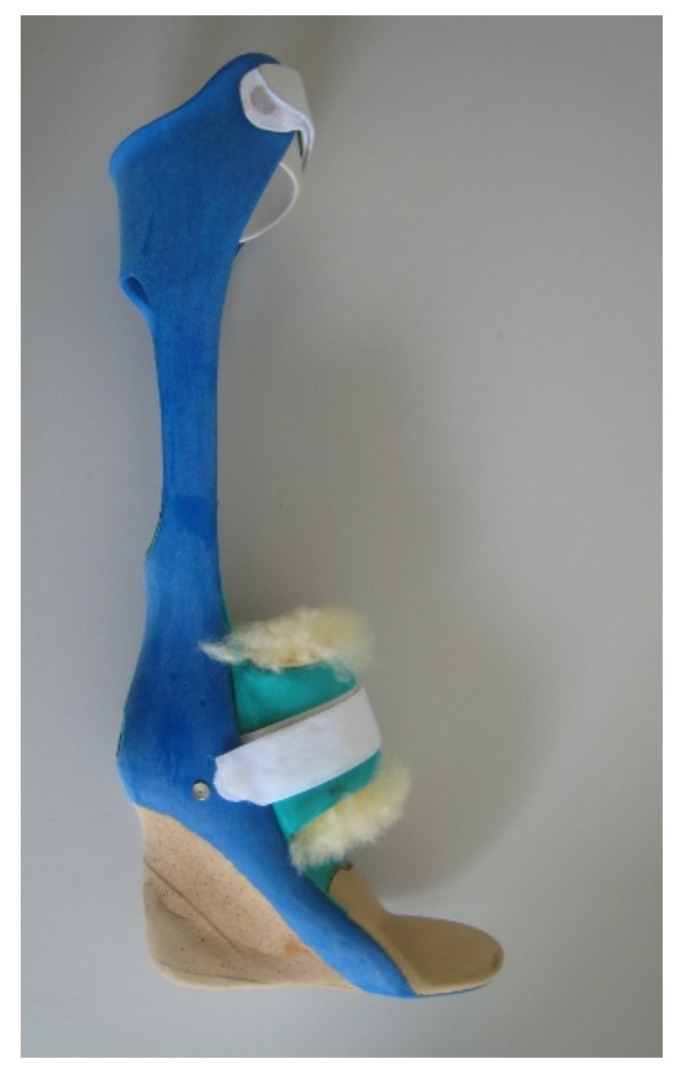
Example of an ankle foot orthosis designed for a severe equinus deformity (rigid version). The added heel shows the compensation for the gastosoleus contracture.

**Figure 3 children-08-00773-f003:**
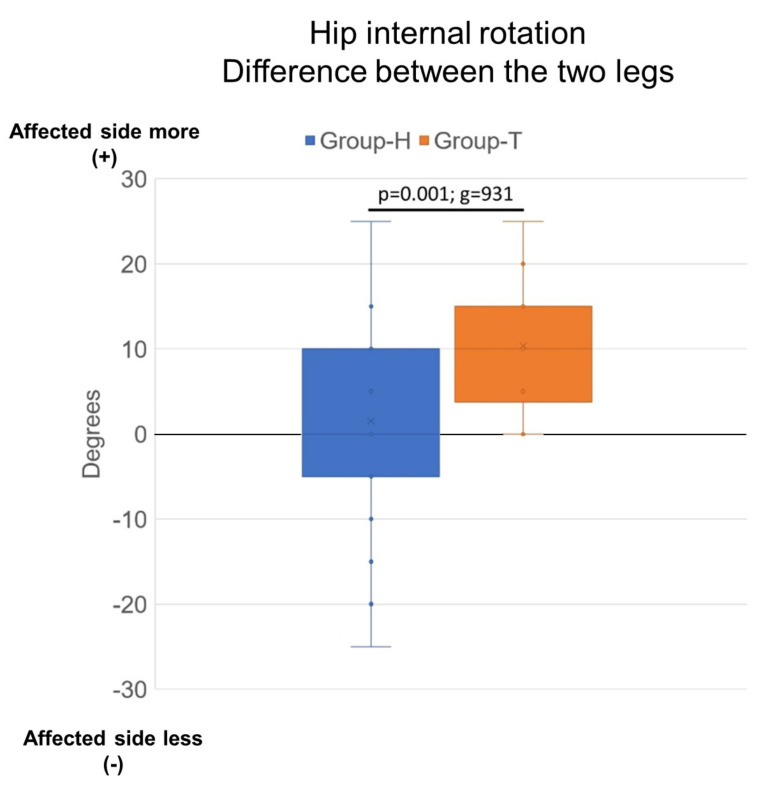
Box plots of hip internal rotation difference between the affected and unaffected sides for both groups (more internal rotation = +); *p* < 0.001, g = 0.93. Group-H: Patients who achieved a heel–toe gait and wore the orthosis regularly about eight hours daily, Group-T: Patients who either rejected the orthosis, wore it only for a short time per day, or walked on their toes despite the orthosis.

**Table 1 children-08-00773-t001:** Description of the patient cohorts and overview of results. Group-H: Patients who achieved a heel-toe gait and wore the orthosis regularly (≥ 8 h a-day), Group-T: Patients who remained toe-walking. They either rejected the orthosis, wore it only for a short time per day (< 8 h per day), or walked on the toes despite the orthosis, GMFCS: Gross Motor Function Classification Scale, AFO: Ankle Foot Orthosis, FO: Foot Orthosis, IR: Internal Rotation. As spasticity (modified Ashworth) was only measured as part of the clinical exam prior to gait analysis, the total number is lower than the total number of the group. Results reported as mean [min, max]. Negative values for the maximum dorsiflexion indicate an equines deformity.

	Group-H	Group-T	*p*-Value	Hedges’g
N	56	14		
Sex (m/f)	34/22	9/5		
Age (years)	11.8 [4.1–32.7]	10.4 [4.0–20.0]	0.352	
Orthosis (stiff AFO/flexible AFO/FO)	3/49/4	0/12/2		
GMFCS (level I/level II)	54/2	13/1		
Spasticity (0/1/2) Ankle Knee	21/31/2 45/9/0	5/5/1 10/1/0		
Max. dorsiflexion	−2.0 [−30, 20]	−5.4 [−30.0, 10.0]	0.527	
Difference in hip IR (+ = more on affected side)	1.5 [−25.0, 25.0]	10.4 [0.0, 25.0]	**0.001**	**0.931**

## Data Availability

The hip internal rotation data presented in this study are available in Appendix A. Further data presented in this study are available on request from the corresponding author.

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
