# Peer review of "Restoration of Heel–Toe Gait Patterns for the Prevention of Asymmetrical Hip Internal Rotation in Patients with Unilateral Spastic Cerebral Palsy"

_children, 2021, doi:10.3390/children8090773_

Round 1

Reviewer 1 Report

The authors present an article about the impact of spastic equinus foot in children with unilateral spastic cerebral palsy and its effect on the function of the proximal lower limb with increased internal rotation of the hip and pelvic retraction. They postulate, that by achieving a heel-to-toe gait pattern through orthotic therapy, ignoring a possibly underlying structural equinus foot, is beneficial for a symmetrical hip rotation between the affected and unaffected leg and therefore prevents the need of future femoral correction surgeries in adolescence other than equinus foot correction.

Abstract:

Line 19: How was the difference of hip internal rotation with the hip in extension assessed, clinically or functionally with gait analysis?

Line 27/28: Do you mean a secondary functional deformity or an actual fixed deformity / bony deformity?

Introduction:

Line 36/37: I would suggest replacing the term “normal” to something less value laden and more neutral such as “physiological”.

Line 37: Despite we all know it, please provide explanation of this abbreviation nonetheless.

Line 42: See Line 36/37. I would suggest replacing the term “normal” to something less value laden and more neutral such as “physiological”. I.e. “the affected feet initially show rather good function during gait, but tend to develop a toe-walking pattern within month or years.” Or similar.

Line 44: Please add: And physiotherapy.

Line 45: Please add: Injections

Line 52: Spasticity in proximal levels was not found in your cohort or in the literature? Maybe change to: “is typically not found”.

Materials and Methods:

Lines 89-101: Wrong section. If this information relates to the actual findings – Results, if it is general information supporting the research question – Introduction.

Line 123: When was gait analysis done? Before start of the orthotic treatment? How long before? Was it used for follow-up as well? Did the patients show similar gait patterns in the sagittal plane as well or were there patients with recurvatum gait and patients with crouch gait? Were parameters of the gait analysis considered for treatment decisions or why were only patients with gait analysis included in this study, but only clinical parameters are measured?

Table 1: How were the ranges of motion evaluated? Clinically? With an extended knee? Do I understand the range correctly, that there were patients without a structural equinus foot at all? Did they have a dynamic equinus gait?

There are several questions of importance to be answered:

Although this is a very interesting and important topic, a clear study design remains unclear to me. Which treatment decisions were made within the clinical routine of the authors, which due to the study? How exactly were the groups defined and on which findings, and after what time of treatment? How long was the follow-up? Subgroups would be interesting in terms of the sagittal gait profile (recurvatum vs crouch). Gait analysis parameters before and after treatment such as plantar flexion moments, pelvic rotation, mean hip rotation in stance, knee extension and knee extension moments would be of high interest.

The thought that achieving a heel-to-toe gait pattern corrects internal rotation gait intriguing, but not supported by any of the findings mentioned. The thought that this prevents pathological bone growth and therefore the need of future femoral osteotomies is even more intriguing. This would have to proofed in a randomized controlled trials with follow-ups over a reasonable time of growth obviously.

Reviewer 2 Report

First, this is an interesting paper. However, as currently written I am not certain I understand the premise of the paper.

Are the authors trying to say if a person has a heel-toe gait pattern with their orthosis, then they will have less asymmetrical passive hip internal rotation?

This is based on the premise, that with time, if the patient's walked with less hip internal rotation, then less asymmetry of passive hip internal rotation would occur?

If the answers to both questions are 'yes', then I do not believe the premises are supported by the data.

There are too many potential confounding factors that cannot be controlled for that could affect passive hip internal rotation therefore conclusions cannot be supported by the data presented.

I do not believe it is appropriate to say that because someone walks with a heel-toe gait pattern that that person will therefore demonstrate less asymmetrical passive hip internal rotation. Additionally, although there is a significant difference between the two means of difference in hip internal rotation. I do not believe a causation can be established with the data presented.

Why?

  1. Your subjects age ranges from 3 to 32 years old. Therefore, unless all of your patients have worn their AFO throughout their growth and development, then there are numerous other factors that could affect passive hip internal rotation. These factors include growth, skeletal maturity, radiographic abnormalities, etc.
  2. It is not clear how long subjects have been wearing the orthosis. How long has each subject worn their AFO? Months? Years? IF years, have they always demonstrated a heel toe gait pattern?
  3. What type of gait study was performed (visual, 3-d)? If 3-d whole body gait studies were performed, then reporting differences  of hip internal rotation while walking would support your theory that over time, if a patient walks with less asymmetrical hip rotation, then they may have less passive hip internal rotation asymmetry.
  4. However, there are too many factors that could affect/contribute to passive hip internal rotation asymmetry than just an appropriate fitting AFO.

Reviewer 3 Report

Thank you for submitting this paper to Children. The manuscript under consideration: "A heel-toe gait pattern leads to symmetrical hip internal rotation in patients with unilateral spastic cerebral palsy" is an interesting article on an important topic in Children. However, there are a few concerns.
1. In the study  patients with CP participated. How did the authors determine the sample appropriate size?
 2. Why did you exclude children with bilateral spastic cerebral palsy? This data could be of interest to understand the motor pattern even in these patients.
3. If available, add more specific information on intelligent quotient. 
4. Give more information on the age of the sample, why it was chosen, and give some context for results of neurologic examination at this age.
5. Table 1 is not looking good. Please add the correct result and fix the table.

Round 2

Reviewer 1 Report

Dear authors

I would like to thank you for the promt altering of the manuscript. The changes made the data and what was done much clearer for me, thank you.

I very much agree on communicating about this important and interesting subject, as it would have a major impact on treating patients with unilateral CP if we would understand the pathology better.

Because of the lack of pre- / post treatment data I still have problems with the conclusions you draw. Patients with a heel-toe gait pattern have symmetrical passive hip rotation and toe walkers don't. The conclusion is, that restoring a heel-toe pattern prevents increased internal hip rotation. But it might be the other way around, as we don't have clinical or instrumental data pre treatment. Maybe patients with symmetrical hip rotation are able to do antegrade foot rocking, while an increased internal hip rotation leeds to increased plantarflexion moments.

Your data very much supports the research question, but in my opinion it does not support the conclusions drawn. The discussion has to dig a little deeper and put the data into perspective.

Reviewer 2 Report

First, I appreciate the amount of time and work that has occurred between the first and second revision of this communication manuscript.

I believe you have improved the manuscript. I also believe the premise has been more clearly stated.

However, in the abstract the following is written: "The aim of this preliminary retrospective cohort investigation was to evaluate the effect of restoring heel-toe gait, through use of functional orthoses, on passive hip internal rotation."

To assess this aim, there must be data before and after orthosis intervention. There is no data demonstrating there is a change in passive hip internal rotation over time.

Additionally, instead of reporting the difference in passive hip internal rotation between involved and uninvolved limbs, perhaps the passive hip internal and external rotation of each side should be reported.

This was written in the conclusion of the communication: "With this communication, we highlighted that a heel-toe gait pattern with anterograde foot rocking, achieved through use of functional orthosis, might lead to more symmetrical clinically assessed passive hip rotation in patients with spastic unilateral cerebral palsy; while toe-walkers with retrograde rocking developed an asymmetrical passive  rotation."

1) there is  no  data demonstrating passive hip internal rotation changed. The difference of hip internal rotation between the two groups could be coincidental. To support the aim of the communication, you would need to present hip ROM before orthosis and hip ROM after orthosis use.

2) Femoral anteversion does not worsen over time, instead it is believed the excessive internal rotation does not decrease over time. (Journal of Anatomy. Scorcelletti et al. 2020; 237:811-826, Annals of  Rehabilitation Medicine. Kong et al. 2018; 42(1):137-144)
